# Conditional Survival in Patients with Locally Advanced Rectal Cancer and Pathologic Complete Response: Results from an Observational Retrospective Multicenter Long-Term Follow-Up Study

**DOI:** 10.3390/cancers17162707

**Published:** 2025-08-20

**Authors:** Carlos Cerdán Santacruz, Oscar Cano-Valderrama, Laura Melina Fernández, Ramón Sanz-Ongil, Rocío Santos Rancaño, Miquel Kraft Carre, Francisco Blanco Antona, Inés Aldrey Cao, Alba Correa Bonito, Jesús Cifuentes, Antoni Codina-Cazador, Eloy Espín-Basany, Eduardo García-Granero, Blas Flor Lorente

**Affiliations:** 1Hospital Universitario de la Princesa, 28006 Madrid, Spain; carlos.cerdan@salud.madrid.org (C.C.S.);; 2Complejo Hospitalario Universitario de Vigo, 36312 Vigo, Spain; oscar.cano.valderrama@sergas.es; 3Champalimaud Foundation, 1400038 Lisbon, Portugal; 4Hospital Central de la Cruz Roja San José y Santa Adela, 28003 Madrid, Spain; 5Hospital Nuestra Señora del Prado, 45600 Talavera de la Reina, Spain; 6Hospital Vall D’Hebron, 08035 Barcelona, Spain; miquel.kraft@vallhebron.cat (M.K.C.);; 7Complejo Asistencial Universitario de Salamanca, 37007 Salamanca, Spain; fblancoa@saludcastillayleon.es; 8Complexo Hospitalario Universitario Ourense, 32005 Ourense, Spain; 9Complejo Hospitalario Universitario de Albacete, 02008 Albacete, Spain; 10Hospital Universitario Doctor Josep Trueta, 17007 Girona, Spain; 11Hospital Universitario y Politécnico de La Fe, 46026 Valencia, Spainflor_bla@gva.es (B.F.L.)

**Keywords:** locally advanced rectal cancer, neoadjuvant therapy, total mesorectal excision, pathologic complete response, conditional survival

## Abstract

Patients with locally advanced rectal cancer who achieve complete remission after pre-surgical chemo-radiotherapy often have a better prognosis than other groups, yet their follow-up care remains the same, leading to unnecessary psychological and financial burdens. This study examines survival trends in these patients using data from multiple Spanish hospitals to determine whether follow-up protocols could be adapted based on conditional survival rates. By analyzing recurrence-free survival over time, researchers found that the probability of remaining cancer-free increases as patients pass key time markers post-treatment, with recurrence rates dropping significantly after three years. These findings suggest that follow-up strategies could be personalized to reduce unnecessary interventions and improve patient well-being, potentially influencing future cancer management guidelines.

## 1. Introduction

The follow-up of colorectal cancer remains a topic of ongoing debate among many authors, with no unanimous consensus on its true contribution to patient outcomes or the most favorable follow-up regimen. This lack of agreement leads to significant variability between centers and even among physicians within the same center, despite similar patient profiles. Although more exhaustive follow-up schedules have not shown clear advantages in terms of overall survival and disease-free survival [1,2,3,4], the general tendency is to utilize them [5,6].

This convention is based on the belief that the progressive acquisition of information will generate sufficient knowledge to design improved follow-up programs, which could provide oncological benefits. Currently, it is essential to consider the direct impact of these follow-ups on two specific and highly sensitive aspects: first, the psychological impact on patients [6,7,8,9], who may feel vulnerable and experience significant stress and anxiety both in the lead-up to tests and while awaiting the final results; and second, the economic burden of the disease [10] in a context where survival rates are increasing, and where the incidence of colorectal cancer is expected to increase among younger populations [11,12,13].

It seems clear that we can leverage the opportunity to acquire meaningful clinical information and apply different approaches to the data at our disposal for individualized patient monitoring, maximizing efficiency. Traditionally, the risk of oncologic adverse events has been established based on postoperative staging and response to neoadjuvant treatment, treating it as a static measure. However, since most local and distant recurrences occur within the first two years post-treatment, this risk is actually a dynamic concept, allowing for modulation of follow-up schedules. This phenomenon is effectively illustrated by conditional survival models, which have thus far been primarily applied to patients LARC undergoing watch-and-wait strategies after achieving a cCR following nCRT [14,15]. Unlike traditional survival models, such as Kaplan–Meier or Cox regression, which estimate survival from a fixed point and yield cumulative probabilities regardless of a patient’s recurrence-free interval, CS models provide dynamically updated survival probabilities based on the time already spent disease-free [16]. This makes them particularly valuable in the context of pCR after nCRT, where the risk of recurrence diminishes over time, allowing for a more nuanced and individualized assessment of prognosis that can better inform follow-up strategies and patient counseling

Among rectal cancer patients, those operated on with curative intent for LARC who have achieved a pathological complete response (pCR) to NCRT are in a more advantageous position due to their optimal oncological results [17,18]. Based on this premise, we have analyzed the survival of these patients using conditional probability, with the aim of obtaining meaningful results that could provide more precise recommendations to improve their follow-up.

## 2. Materials and Methods

This study is a secondary analysis of an original study registered at ClinicalTrials.gov, number NCT05495308. In that study, the oncological outcome of a highly selected group of rectal cancer patients, those with a pCR after neoadjuvant treatment regardless of their basal clinical staging, was investigated [17].

### 2.1. Study Design and Population

This multicenter observational study was based on prospectively collected data from the Spanish Rectal Cancer Project Registry.

In 2006, the AEC launched an audited educational initiative modeled on the Norwegian Colorectal Cancer Project [19], aiming to improve rectal cancer treatment outcomes nationwide. This was achieved by introducing TME techniques to multidisciplinary teams across a network of participating hospitals. The registry remained active until its formal closure in November 2017. Data integrity was ensured through audits by both the program coordinator and relevant health authorities. Detailed accounts of the initiative have been published previously [20,21]. Institutions joined on a voluntary basis, submitting data on consecutive rectal cancer patients to a secure online platform. Neither researchers nor hospitals received financial compensation for their contributions.

For the current study, only institutions with more than ten pCR cases documented in the registry were invited to participate.

The design and reporting followed the STROBE Statement for observational studies [22].

Inclusion and exclusion criteria: Eligible patients were aged over 18, had undergone surgery with curative intent, and received neoadjuvant therapy. Both TME and PME procedures were included. Cases involving local excision, palliative surgery, or synchronous metastases at diagnosis were excluded. Additionally, any patient with fewer than six months of follow-up was excluded from oncological outcome analysis.

### 2.2. Definitions and Variables of Interest

Rectal cancer was defined as any tumor located within 15 cm of the anal verge, as measured by MRI or rigid rectoscopy.

Data collection was conducted by senior medical staff at each participating center. Variables included demographic information, preoperative and operative details, pathological findings, hospital admission data, 30-day postoperative outcomes, and oncologic results. Pathology classification followed the 8th Edition of the TNM system [23]. Follow-up was prospective until 2017 when the registry was interrupted. In the group of patients with pCR, considered as the group for investigation, an updated follow-up was performed in December 2021.

Local recurrence and distant metastases were defined a priori in the study protocol as follows: presence of any oncological disease, both radiologically suspected and with pathological confirmation. Local recurrence was defined as any recurrence within a previously irradiated field, including the region of the anastomosis, presacral region, and lateral pelvic nodes. Any other form of disease relapse in lymph nodes in territories other than those typically found in the radiotherapy field, peritoneum, or other organs was considered distant metastasis. Diagnosis was based on radiological studies, such as thorax or abdomen CT, pelvic MRI or PET-CT, and/or histological samples, both percutaneously and surgically acquired samples.

### 2.3. Treatment Regimens and Follow-Up

Neoadjuvant treatment indications, adjuvant chemotherapy regimens, and follow-up schedules were determined according to the best clinical practices and current international guidelines. Decisions were tailored to individual patient characteristics and preferences, following multidisciplinary team (MDT) discussions. Completion of adjuvant chemotherapy was defined as administration of at least 80% of the originally prescribed total dose.

Multiple neoadjuvant strategies were documented, primarily including:•Long-course chemoradiotherapy: Radiotherapy with a total dose of 50.4 Gy delivered over 20–25 fractions during a five- to six-week period, combined with capecitabine-based chemotherapy.•Short-course radiotherapy: Radiation alone or with chemotherapy, totaling 28 Gy in five fractions over one to two weeks.•Preoperative chemotherapy alone, without concurrent radiotherapy.

Follow-up data were updated specifically for this study, with December 2021 recorded as the last follow-up date.

### 2.4. Aims and Outcome Measures

In the present work, the main outcome measure was long-term recurrence (both local and systemic) in the selected population of LARC patients with pCR after NCRT and TME or PME surgery.

Recurrence was studied as recurrence-free survival (RFS), which was assessed based on events defined as death or recurrence of any type, local or systemic, of the patient; and as overall recurrence, which was assessed based on events defined as recurrence of any type, local or systemic. In this case death was censored and was not considered an event. RFS was studied using a survival function estimated by the Kaplan–Meier method, while global recurrence was studied using a failure function estimated by the same method.

We used conditional survival modeling to estimate the probability of patients remaining alive and free of disease annually for those patients who were survivors at one, two, and three years after the procedure.

### 2.5. Statistical Method

Qualitative variables are presented with their frequency distribution. Quantitative variables are represented by their mean and standard deviation (SD) or median and interquartile range (IQR) in case of asymmetry.

Univariable analysis was performed with a Cox proportional hazard model to assess the association between the different independent variables and RFS. In order to adjust for confounding factors, a multivariable analysis was performed using a Cox proportional hazard model. Variables that had a statistically significant association (*p* < 0.1) or clinical relevance [Hazard ratio (HR) > 1.5 or HR < 0.67] in the univariable analysis were included in the multivariable analysis. The selection of the definitive model was carried out using the forward stepwise method with an inclusion value in the model of *p* <0.05 and exclusion of *p* > 0.10. *p* < 0.05 was considered to indicate statistical significance (two-tailed test).

Time to recurrence (local recurrence and distant metastasis) was used to create survival curves of RFS. Conditional recurrence-free survival (absence of local or distant recurrence) analyses were subsequently used to investigate changes in the probability of recurrence as patients remained recurrence-free after surgery and to assess possible changes in the effect of prognostic factors over time.

All analyses were performed using Stata 13.1 (StataCorp, College Station, TX, USA).

## 3. Results

The study involved 32 Spanish hospitals, which collectively contributed to the Vikingo Project Registry. Between March 2006 and November 2017, the registry compiled data from a total of 12,082 patients.

Following the application of predefined inclusion and exclusion criteria, a final cohort of 815 patients was selected for analysis (Figure 1).

### 3.1. Patients’ Characteristics

The mean age of the study population was 65.1 years (95% CI 64.4–65.9), and 294 (36.1%) of the patients were female. Most patients were classified according to the American Society of Anesthesiologists (ASA) classification as ASA I or II (534, 65.5%), and the most frequent tumor location was the middle rectum (411 patients, 50.4%). The most frequently used surgical techniques were anterior rectal resection (635 patients, 78.9%) and abdominoperineal resection (170 patients, 21.1%). The surgical approach was laparoscopic in 474 (58.2%) patients. Of the total number of patients included, 300 (36.8%) developed some kind of complication during the postoperative period, with 65 (8.0%) of them requiring red blood cell transfusion and 60 (7.4%) requiring reoperation. The quality of the removed mesorectum was satisfactory in 683 (85.3%) patients. After hospital discharge, 595 (73.0%) patients underwent adjuvant treatment.

### 3.2. Oncologic Outcomes

At the end of flow-up (median follow-up time of 73.4 months, IQR 57.7–99.5), we observed recurrence in 61 individuals (7.48%), categorized as local (15 patients, 1.84%) or distant (52 patients, 6.38%). The median time to recurrence was 17.6 months (IQR 10.4–28.0). Among these, 68% had received adjuvant therapy, and statistical analysis did not reveal significant differences between those who did and did not recur (*p* > 0.91). Global recurrence was 7.27% in patients who did not receive adjuvant therapy and 7.12% in those who did (*p* = 0.917). Local recurrence rates were 1.82% and 1.85% (*p* = 0.964), and distant recurrence occurred in 5.91% vs. 5.98% of patients, respectively (*p* = 0.956). Although recurrence rates were similar between groups, the absence of adjuvant therapy was associated with a significantly increased risk of death, with an HR for OS of 1.6 (95% CI: 1.1–2.3; *p* = 0.01)

Local recurrences were distributed anatomically as follows: anterior (20%), axial (40%), lateral (20%), and posterior (20%). Distant recurrences were most frequently hepatic (30.8%), followed by pulmonary or osseous (5.8%), peritoneal (3.9%), cerebral (1.9%), and other sites (9.6%). Although post-recurrence interventions were not uniformly recorded, a subset of patients underwent further treatment following disease recurrence. Median time to recurrence varied by type: 27.8 months [95% CI: 16.1–71.9] for local recurrences, 17.3 months [95% CI: 12.9–22.4] for distant recurrences, and 13.8 months [95% CI: 6.9–21.4] for the global group.

Regarding survival status, 60% of patients with local recurrence had died, with 46.7% of these deaths attributed to cancer. For distant recurrences, 73.1% of patients had died, with 55.8% cancer-related. In the global recurrence group, 70.5% had died, of which 54.1% were due to cancer.

The impact of surgical procedure and approach on overall survival was evaluated. Patients who underwent AR had a mortality rate of 15.1% (96/635), compared to 22.9% (39/170) in the APR group. The HR for OS in APR versus AR was 1.5 (95% CI: 1.1–2.2; *p* = 0.03), indicating a significantly worse outcome associated with APR. Regarding the surgical approach, 13.9% of patients (66/474) treated with MIS died during follow-up, versus 20.0% (68/341) in the OS group. The HR for OS in OS versus MIS was 1.3 (95% CI: 0.9–1.8; *p* = 0.21), showing a non-significant trend favoring MIS.

The estimated RFS rates of the study population at 12, 24, 36, 48, 60, 72, 84, and 96 months were 96.8%, 93.2%, 90.9%, 88.6%, 86.5%, 84.3%, 82.7%, and 79.8%, respectively (Figure 2). The association of different sociodemographic and clinical characteristics of this population with the development of tumor recurrence was studied in a univariable analysis (Table 1).

### 3.3. Conditional Disease-Free Survival

Of the 815 patients, 35 died or experienced recurrence in the first postoperative year; therefore, 780 patients (one-year survivors) were included in the conditional survival analysis at one year after surgery. Of these patients, 42 (5.4%) experienced some form of recurrence during the follow-up. The RFS of the overall population and that of one-year survivors throughout the follow-up are shown in both Figure 2 and Table 2. The multivariable analysis in one-year survivors showed that age (HR 1.06, 95% CI 1.04–1.08, *p* < 0.001) and ASA classification III or IV (HR 1.8, 95% CI 1.3–2.7, *p* = 0.001) were the only variables significantly associated with RFS at one year after the intervention.

Two years after the intervention, 745 patients remained alive and free of recurrence (two-year survivors). In this group, 22 (3.0%) patients experienced recurrence. The evolution of RFS in these patients throughout the follow-up is shown in Figure 2 and Table 2. In this group of patients, the multivariable Cox model showed that age (HR 1.07, 95% CI 1.05–1.09, *p* < 0.001) and ASA classification (HR 2.0, 95% CI 1.3–3.0, *p* = 0.002) were the only variables significantly associated with RFS at two years after the intervention.

Three years after surgery, 720 patients remained free of recurrence (three-year survivors). In this group the recurrence rate was 1.8% (13 patients). The evolution of RFS in these patients throughout the follow-up is shown in Figure 2 and Table 2. In the multivariable analysis in this group, the only factors significantly associated with RFS at three years after the intervention were age (HR 1.08, 95% CI 1.04–1.10, *p* < 0.001) and ASA III or IV (HR 1.8, 95% CI 1.1–2.9, *p* = 0.017).

The evolution of recurrence in the overall population and in one-, two-, and three-year survivors is shown in Figure 3 and Table 3. Altogether, these results show that the probability of RFS increases in LARC patients with pCR after NCRT followed by surgery as they progress through the follow-up without recurrence.

## 4. Discussion

Patients with LARC treated with NCRT and who undergo surgery with a pCR in the surgical specimen can be considered a privileged group, with an excellent prognosis both locally and in terms of distant metastases, high overall survival, and recurrence-free survival [17,18,24].

The present work also shows that in this group of patients, as time passes, those who survive progressively increase their conditional probability of overall and disease-free survival in the next control time interval. Although this result may seem obvious, it has not been previously confirmed by any research, and we therefore consider it as important information.

The same behavior has been previously proven in a similar clinical setting, in LARC patients who underwent NCRT with complete clinical response and were managed with a watch-and-wait strategy [14,15].

Even if the role of adjuvant chemotherapy in patients with pCR is still debated, our data did not show meaningful differences in recurrence. All *p*-values that compare the differences between patients who did and did not receive adjuvant therapy were far from significant. These results suggest that in this subgroup, adjuvant chemotherapy may not provide clear benefit, which makes sense considering their good prognosis. This interpretation should also be viewed in the context of evolving treatment paradigms, as emerging strategies, such as total neoadjuvant therapy, may influence future outcomes and recommendations [25,26].

In practical terms, the results of this work could be used in several ways. They offer the opportunity to provide more accurate information to patients and their families about survival expectations in a scenario that, to date, has been assumed to be highly advantageous with little scientific evidence. In addition, having better quality information could be beneficial for patients by helping them to reduce the levels of anxiety and stress caused by successive follow-up visits. Moreover, anxiety and stress could be further reduced by the potential reduction in the number of diagnostic tests based on the predictable absence of significant findings.

Moreover, our results could shed light on the controversy regarding the efficiency offered by exhaustive follow-up programs [1,3,4,15]. For instance, surveillance guidelines from the American Society of Colon and Rectal Surgeons [5] and the American Cancer Society [27] offer distinct follow-up recommendations based on cancer stage. For stage I, no routine follow-up is advised. Patients with stages II–III are typically monitored through clinical visits, CEA testing, CT imaging, and colonoscopy until year 5, while stage IV patients treated with curative intent require more intensive surveillance.

Notably, these guidelines do not include specific recommendations for patients with pCR after nCRT, highlighting the need for individualized follow-up in this subgroup. Considering the favorable prognosis and declining recurrence risk among pCR patients, a reduction in surveillance duration and intensity may be appropriate. Although still debated, a structured three-year follow-up could provide a balanced alternative to the standard five-year protocol. A different schedule for this group—guided by their particularly good prognosis and supported by conditional survival analysis—could help fill the gap in current international recommendations.

It also has to be taken into account that, as total neoadjuvant treatment strategies gain increasing interest and clinical application, all these figures and their interpretations are likely to change, requiring new and adapted recommendations in that context [25,26].

The results of the present study support the use of much more lax follow-ups managed by general practitioners or advanced practice nurses, which would allow for optimizing healthcare resources and reducing costs.

Unfortunately, apart from this information, a multivariable analysis using logistic regression did not identify potential clinical or patient-associated factors capable of predicting the occurrence of adverse oncological events during follow-up. These results are in accordance with those of a previous study [17] with a more global view and without conditional probability analysis. Although our study population was large (815 patients), the low incidence of recurrence in these patients has likely hampered the identification of these predictors with the conventional statistical models available to us. Moreover, due to the low number of recurrence events observed in the cohort, statistical power in multivariable analysis is limited, which may reduce the ability to identify rare but potentially relevant prognostic factors.

Nevertheless, some clinical variables did emerge as relevant. Age and ASA score were identified as independent predictors of RFS and should therefore be taken into account when designing follow-up strategies for these patients. International guidelines already recognize certain risk factors, which compel clinicians to adapt surveillance models by increasing the frequency or scope of monitoring in higher-risk cases. Likewise, it is appropriate to personalize follow-up schedules based on individual risk profiles, enhancing efficiency while minimizing unnecessary interventions in patients with lower risk.

This constraint is inherent to studies addressing infrequent clinical outcomes.

This study has some limitations, such as the retrospective update of part of the long-term survival data and the variability of treatment between centers and different time periods throughout the entire audit project. In addition, as participation in the project was voluntary, there is an inherent risk of selection bias that could not be eliminated. Another limitation is the lack of genetic information from the diagnostic biopsy, although the impact of this limitation may be small because the large intratumoral variability could preclude us from obtaining consistent findings [28,29,30]. Moreover, the impact of MMR or MSI status on survival outcomes in patients who achieved pathological complete response (pCR) following neoadjuvant chemoradiotherapy (nCRT) remains unclear. A recent systematic review and meta-analysis published in *Cancers* (2023) [29] concluded that MMR/MSI is not significantly associated with pCR rates after nCRT and found no significant differences in response between MSI and MSS tumors.

As stated above, this study presents a global analysis of conditional survival, offering a novel perspective on long-term outcomes in patients with pCR after nCRT. This approach provides dynamic prognostic information over time and may help refine follow-up strategies beyond traditional static survival estimates [15,16]. However, while conditional survival represents an important step forward in understanding prognosis, future analyses should aim to incorporate specific factors, such as adjuvant therapy and histopathological risk features, which may further individualize surveillance. Due to the low frequency of adverse events and the wide range of variables that could influence prognosis, such detailed stratification was not feasible within the current cohort. Larger datasets will be necessary to explore these associations with adequate statistical power and to enhance the precision of personalized follow-up models.

On the other hand, this study also has important strengths, such as its multicenter nationwide nature, with one of the largest populations of LARC patients with pCR after NCRT, and the application of conditional survival concepts, which offers additional information to a rarely explored field.

Several practical implications can be drawn from the current study. For example, based on our data, more personalized follow-up strategies could be established and adapted to the different patient specific follow-up times, which is not yet fully accepted by some medical societies. In addition, this information could increase the decision-making capacity of patients, which could contribute to increase the shared decision-making process that is increasingly advocated in modern medicine.

## 5. Conclusions

The results of this study suggest that the frequency and intensity of active surveillance can be safely modified in patients with LARC treated with neoadjuvant chemo-radiotherapy and subsequent surgery who achieve pCR. A follow-up longer than three years can be considered inefficient regarding distant and local recurrences, as the proportion of patients who might develop any kind of recurrence is very low.

## Figures and Tables

**Figure 1 cancers-17-02707-f001:**
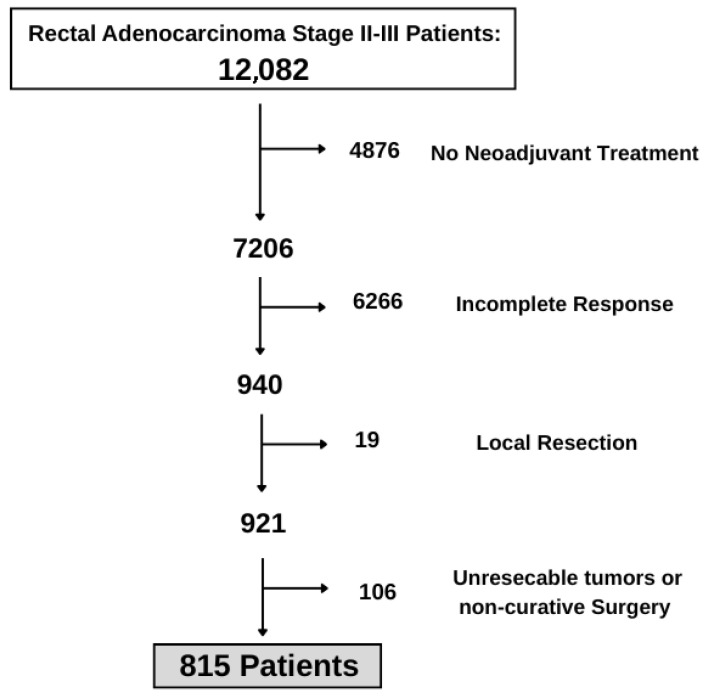
Flowchart detailing patient selection for the present study.

**Figure 2 cancers-17-02707-f002:**
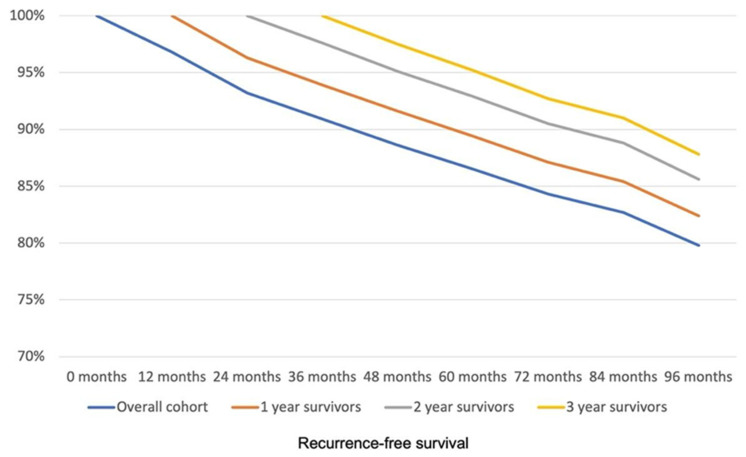
Evolution of recurrence-free survival throughout the follow-up. The graph represents recurrence-free survival for the study population (overall cohort), and for patients without recurrence at one year (one-year survivors), two years (two-year survivors), and three years (three-year survivors) of follow-up.

**Figure 3 cancers-17-02707-f003:**
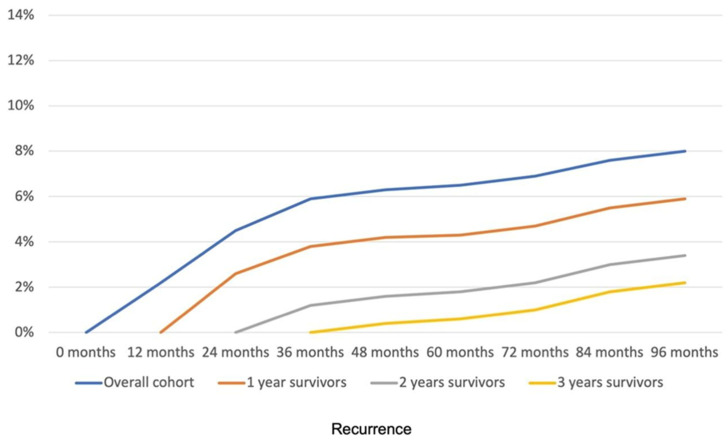
Evolution of recurrence throughout the follow-up. The graph represents recurrence for the study population (overall cohort), and for patients without recurrence at one year (one-year survivors), two years (two-year survivors), and three years (three-year survivors) of follow-up.

**Table 1 cancers-17-02707-t001:** Univariable analysis of the main factors related to the development of tumor recurrence.

	No Recurrence (n = 754)	Recurrence (n = 61)	HR (RFS)	*p*
Gender (male)	486 (64.5%)	35 (57.4%)	1.2 (95%CI 0.8–1.6)	0.385
Age (years)	65.1	66.1	1.06 (95%CI 1.05–1.08)	<0.001
ASA * (ASA III o IV)	258 (34.2%)	23 (37.7%)	2.3 (95%CI 1.7–3.2)	<0.001
Low rectal cancer	266 (35.3%	26 (42.6%)	-	0.506
CEA < 5 ng/mL	106 (15.0%)	15 (28.3%)	0.7 (95%CI 0.5–1.1)	0.084
APR	21 (34.4%)	149 (20.0%)	1.4 (95%CI 1.0–2.0)	0.053
TME Surgery	675 (90.6%)	54 (88.5%)	0.9 (95%CI 0.5–1.5)	0.744
Perforated tumor	20 (2.7%)	4 (6.6%)	2.0 (95%CI 0.9–4.2)	0.081
Open surgery or conversion	312 (41.4%)	28 (45.9%)	1.3 (95%CI 0.9–1.8)	0.121
Complications	274 (36.3%)	26 (42.6%)	1.4 (95%CI 1.0–1.9)	0.051
Blood transfusion	58 (7.7%)	7 (11.5%)	1.7 (95%CI 1.1–2.7)	0.018
Adjuvant chemotherapy	550 (72.9%)	45 (73.8%)	0.7 (95%CI 0.5–0.9)	0.013
Satisfactory mesorectum	632 (85.4%)	51 (83.6%)	-	0.841

* ASA: American Society of Anesthesiologists classification; CEA: carcinoembryonic antigen; CI: confidence interval; APR abdominoperineal resection; TME: total mesorectal excision; HR: hazard ratio; RFS: recurrence-free survival).

**Table 2 cancers-17-02707-t002:** Evolution data of recurrence-free survival throughout the follow-up. Data represent recurrence-free survival at different time points for the study population (overall cohort), and for patients without recurrence at one year (one-year survivors), two years (two-year survivors), and three years (three-year survivors) of follow-up.

	Global Cohort	One-Year Survivors	Two-Year Survivors	Three-Year Survivors
0 months	100.0%	-	-	-
12 months	96.8%	100.0%	-	-
24 months	93.2%	96.3%	100.0%	-
36 months	90.9%	93.9%	97.6%	100.0%
48 months	88.6%	91.6%	95.1%	97.5%
60 months	86.5%	89.4%	92.9%	95.2%
72 months	84.3%	87.1%	90.5%	92.7%
84 months	82.7%	85.4%	88.8%	91.0%
96 months	79.8%	82.4%	85.6%	87.8%

**Table 3 cancers-17-02707-t003:** RFS data of the evolution of recurrence throughout the follow-up. Data represent recurrence at different time points for the study population (overall cohort), and for patients without recurrence at one year (one-year survivors), two years (two-year survivors), and three years (three-year survivors) of follow-up.

	Overall Cohort	One-Year Survivors	Two-Year Survivors	Three-Year Survivors
0 months	0.0%	-	-	-
12 months	2.2%	0.0%		-
24 months	4.5%	2.6%	0.0%	-
36 months	5.9%	3.8%	1.2%	0.0%
48 months	6.3%	4.2%	1.6%	0.4%
60 months	6.5%	4.3%	1.8%	0.6%
72 months	6.9%	4.7%	2.2%	1.0%
84 months	7.6%	5.5%	3.0%	1.8%
96 months	8.0%	5.9%	3.4%	2.2%

## Data Availability

The raw data supporting the conclusions of this article will be made available by the authors upon request. Interested researchers should contact Dr. Oscar Cano Valderrama at oscar.cano.valderrama@sergas.es to request access.

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
