# Peer review of "Conditional Survival in Patients with Locally Advanced Rectal Cancer and Pathologic Complete Response: Results from an Observational Retrospective Multicenter Long-Term Follow-Up Study"

_cancers, 2025, doi:10.3390/cancers17162707_

Round 1
Reviewer 1 Report
Comments and Suggestions for Authors
The study analyzed an important issue in the follow-up care of patients with locally advanced rectal cancer who achieve a pathological complete response after neoadjuvant therapy . The topic of personalized follow-up strategies based on conditional survival data could significantly impact clinical practices, optimizing patient care and reducing unnecessary interventions. However, there are some important issues that need to be addressed.
- In the abstract of the paper, the sentence 'The probability of RFS at 5 years was 86.5% in the whole cohort and 89.4%, 92.9% and 95.2% for survivors at one, two and three years, respectively' is incorrectly phrased.
- MMR or MSI status should be included as an analysis factor and added to Table 1.
- The author should provide a more detailed explanation of the conditional survival model and its comparison with traditional survival analysis.
- Conduct a more detailed analysis of recurrent patients, including the site of recurrence, whether they received adjuvant therapy, post-recurrence interventions, and their current survival status. This information is very valuable for guiding the management and treatment of future patients.
- The authors claim that the results may lead to more personalized follow-up strategies. However, considering the existing follow-up care clinical guidelines, they can provide more insights on how to implement these strategies in practice.
This manuscript provides an interesting and timely analysis of conditional survival in rectal cancer patients with pCR after NCRT. These findings are clinically significant and suggest potential changes to follow-up protocols to improve patient care and reduce unnecessary costs. Overall, this study was conducted well, but some additional clarifications and suggestions for further research could enhance its contribution to the field.
Reviewer 2 Report
Comments and Suggestions for Authors
This manuscript presents a retrospective multicentre analysis of 815 patients with locally advanced rectal cancer (LARC) who achieved pathologic complete response (pCR) following neoadjuvant chemo-radiotherapy (NCRT) and surgery. Utilizing conditional survival modeling, the authors demonstrate that the likelihood of remaining recurrence-free improves notably with each passing recurrence-free year, suggesting that follow-up intensity may be safely reduced beyond three years. However, the study design raises some serious concern due to the lack of control group and selection bias. The study also lacks any genetic/molecular data. While the results support a change in follow-up regimen for low-risk groups, it might be counter argumentative for others. I advise major revision of the article with the following questions answered.
- Please enlist the correct protocols for patients with locally advanced rectal cancer (LARC) for follow up checkup and treatment. Also, how they are assessed for cancer? Is there a difference in method of detection in multicenter?
- What is the impact of inter-institutional variation in treatment protocols, follow-up intensity, and reporting? Given the multicentre and long-term nature of the of the treatment can these factors contribute to the outcomes? Please enlist them in a table for readers to consider. Also, how generalizable these findings are for patients outside of Spain?
- Did authors have any data on patient’s background like ethnicity, socio-economic burden?
- Did authors study the effect of the different surgical procedures like anterior rectal resection (635 patients, 78.9%) and 189 abdominoperineal resection (170 patients, 21.1%). laparoscopic 190 in 474 (58.2%) patients on patient survival?
- Can you clarify the rationale for including only centers with >10 pCR patients? How might this criterion have influenced the reported outcomes or generalizability? Did the use or completion of adjuvant chemotherapy in pCR patients impact conditional survival, and could this inform ongoing debate about the value of adjuvant therapy post-pCR?
- Given the low event rate (recurrence), how confident are you in the multivariate analyses regarding prognostic factors as the low event rate for recurrences restricts statistical power, making it difficult to find or validate factors predicting late recurrence or subgroups at different risk? Could rare but important predictors have been missed?
- Did you perform sensitivity analyses to evaluate the robustness of your conditional survival estimates in subgroups, such as different age groups, ASA status, or surgical approaches?
- Do authors have any potential mechanism of late recurrence? If yes please support it with adequate reference.
Reviewer 3 Report
Comments and Suggestions for Authors
This study enrolled 815 patients with locally advanced rectal cancer who yield pCR after neoadjuvant therapy and analyzed the recurrence rate in relation to the duration of follow-up. This is a sizable cohort, and the conclusions drawn are clinically and economically important for the follow-up of this population. The following questions would be appreciated to be addressed by the authors:
- The recurrence patterns in this study included both local and systemic, and I think this should be stated clearly in the abstract.
- What was the follow-up strategy in this study?
- I think that insufficient consideration was given in this study to factors that may affect prognoses, for example, radiological prognostic predictors, including T/N staging, EMVI and MRF status at baseline, which may affect the results of this study to some extent.
- Neoadjuvant therapy regimens were not consistent; how many people were treated with each program? What were the indications for each regimen? Do the different treatment regimens affect patient survival?
- In 3.2. Conditional disease-free survival, the results of the multivariable Cox analysis showed that age and ASA were independent predictors of RFS, what are the potential implications for clinical decision-making, and can this be discussed accordingly?
- It should be uni-/multivariable analyses rather than uni-/multivariate.
Round 2
Reviewer 3 Report
Comments and Suggestions for Authors All my concerns have been addressed.